



# Long-term energy balance measurements at three different mountain permafrost sites in the Swiss Alps

Martin Hoelzle[1], Christian Hauck[1], Tamara Mathys[1], Jeannette Noetzli[2], Cécile Pellet[1], and Martin Scherler[1]

[1]Department of Geosciences, University of Fribourg, Fribourg, Switzerland
[2]WSL Institute for Snow and Avalanche Research SLF, Davos, Switzerland

*Correspondence to:* Martin Hoelzle (martin.hoelzle@unifr.ch)

**Abstract.** The surface energy balance is a key factor influencing the ground thermal regime. With ongoing climate change, it is crucial to understand the interactions of the individual heat fluxes at the surface and within the subsurface layers as well as their relative impacts on permafrost thermal regime. A unique set of high-altitude meteorological measurements has been analysed to determine the energy balance at three mountain permafrost sites in the Swiss Alps (Murtèl-Corvatsch, Schilthorn and Stockhorn), where data is being collected since the late 1990s in collaboration with the Swiss Permafrost Monitoring Network (PERMOS). All stations are equipped with sensors for four-component radiation, air temperature, humidity, wind speed and direction as well as ground temperatures and snow height. The three sites differ considerably in their surface and ground material composition as well as their ground ice contents. The energy fluxes are calculated based on two decades of field measurements. While the determination of the radiation budget and the ground heat flux is comparatively straightforward (by the four-component radiation sensor and thermistor measurements within the boreholes), larger uncertainties exist for the determination of turbulent sensible and latent heat fluxes.

Our results show that mean air temperature at Murtèl-Corvatsch (1997 – 2018, 2600 m asl.) is -1.66 °C and has increased by about 0.7 °C during the measurement period. At the Schilthorn site (1999 – 2018, 2900 m asl.) a mean air temperature of -2.60 °C with a mean increase of 1.0 °C was measured. The Stockhorn site (2003 – 2018, 3400 m asl.) recorded lower air temperatures with a mean of -6.18 °C and an increase of 0.7 °C. Measured net radiation, as the most important energy input at the surface, shows substantial differences with mean values of 30.59 $Wm^{-2}$ for Murtèl-Corvatsch, 32.40 $Wm^{-2}$ for Schilthorn and 6.91 $Wm^{-2}$ for Stockhorn. The calculated turbulent fluxes show values of around 7 to 13 $Wm^{-2}$ using the Bowen ratio method and 3 to 15 $Wm^{-2}$ using the bulk method at all sites. Large differences are observed regarding the energy used for melting of the snow cover: at Schilthorn a value of 8.46 $Wm^{-2}$, at Murtèl-Corvatsch of 4.17 $Wm^{-2}$ and at Stockhorn of 2.26 $Wm^{-2}$ is calculated reflecting the differences in snow height at the three sites.

In general, we found considerable differences in the energy fluxes at the different sites. These differences may help to explain and interpret the causes of the varying reactions of the permafrost thermal regime at the three sites to a warming atmosphere. We recognize a strong relation between the net radiation and the ground heat flux. Our results further demonstrate the importance of long-term monitoring in order to better understand the impacts of changes in the surface energy balance components on the permafrost thermal regime. The dataset presented can be used to improve permafrost modelling studies aiming at e.g. advancing



knowledge about permafrost thaw processes. The data presented and described in this study is available for download at the following site http://dx.doi.org/10.13093/permos-meteo-2021-01 (Hoelzle et al., 2021).

## 1 Introduction

High-mountain regions are particularly vulnerable to climate change (Haeberli and Beniston, 1998; Huggel et al., 2010, 2015).
Especially the alpine cryosphere, including the components snow, glaciers and permafrost, is reacting strongly to the ongoing atmospheric warming (IPCC, 2013). Significant changes in permafrost have already been documented by many observation networks world-wide (GCOS, 2010; Biskaborn et al., 2019; Noetzli et al., 2020). Operational mountain permafrost observations are particularly well established in Switzerland and include the longest mountain permafrost observations world-wide (Hoelzle et al., 2002) and are coordinated within the Swiss Permafrost Monitoring Network (e.g. PERMOS, 2019). As permafrost in
general is not easily detectable from the surface (or space), it is investigated and monitored using temperature measurements in boreholes or non-invasive methods such as near-surface geophysics, which enable the analysis of the subsurface composition as well as the investigation of the ground ice content and its changes. Changes in ground ice content are important because they can be directly related to strong environmental impacts, such as the future development of natural hazard potential as well as changes in runoff amount and patterns, considerably affecting dry mountain regions (e.g. Walvoord and Kurylyk, 2016).
To better understand the complex process chains leading to the above mentioned impacts related to permafrost thaw, it is necessary to improve our knowledge about the thermal regime of permafrost, which is mainly depending on the energy balance at the surface and on the geothermal heat flux at its lower boundary. The energy balance at the ground/snow surface is hereby given by the net radiation balance, the turbulent heat fluxes, the melt energy of the snow cover, the heat flux through the snow cover and the ground heat flux. It has been subject of many field-based observational and modelling studies (e.g. Hoelzle and
Gruber, 2008; Gruber and Hoelzle, 2008; Westermann et al., 2009; Langer et al., 2011a, b; Scherler et al., 2014; Chadburn et al., 2015; Marmy et al., 2016; Pellet et al., 2016). Therefore, it is crucial to better understand the energy balance at the surface in order to improve our understanding of the permafrost thermal regime.

It has to be noted that the permafrost thermal regime may vary substantially as a function of the surface cover. In coarse blocky terrain particular attention has to be given to the ground heat flux, as it can be very different from the mainly conductive
heat flux in soil or bedrock substrates (e.g. Hanson and Hoelzle, 2004; Wicky and Hauck, 2017). Few studies have attempted to model heat transfer processes in such coarse blocky active layers by including e.g. turbulent heat fluxes within the active layer in their models (e.g. Scherler et al., 2014) or, more recently, air convection processes (e.g. Wicky and Hauck, 2017, 2020; Luethi et al., 2017; Pruessner et al., 2018).

However, in contrast to the above cited studies, most permafrost models use much simpler formulations for the ground heat
flux, and are based on estimations of the potential incoming radiation, modified by the effect of snow insulation (Etzelmüller, 2013; Riseborough et al., 2008). Other simplified approaches include thermal offsets which are the differences between air, surface and permafrost temperatures to characterize the influence of snow conditions and thermal conductivities of different





ground materials (Burn and Smith, 1988; Wu et al., 2003; Hoelzle and Gruber, 2008). In more recent permafrost or land surface models (e.g. Fiddes et al., 2015; Ekici et al., 2015; Pellet et al., 2016) the energy balance at the surface is explicitly included.

Nevertheless, there exist few observational datasets of the full energy balance at specific permafrost sites, especially in mountain regions. (Scherler et al., 2014; Boike and Abnizova; Boike et al.; Mohd Wani et al., 2020). As mentioned above, the

energy balance constitutes one of the most important input parameters for the ground heat flux regime, and it is therefore crucial to better understand the influence of the individual fluxes. Thus, such full energy balance datasets are necessary to improve the understanding and the ability to correctly model the various processes for different surface and subsurface materials in permafrost regions.

In this study, we focus on 20 years time series from three permafrost stations in the Swiss Alps (Schilthorn, Stockhorn and

Murtèl-Corvatsch), which were established in the scope of the European research project PACE (Permafrost and Climate in Europe, (Harris et al., 2003)) at the beginning of this century and are now part of the PERMOS network. All stations are equipped with standard meteorological sensors such as a four component radiation sensor, air temperature, humidity, wind speed and direction as well as ground temperatures and snow height (Hoelzle and Gruber, 2008). All three sites differ considerably in their ground material composition and permafrost conditions. The influence of the radiation components and the turbulent heat

fluxes on the ground heat fluxes and the snow melt energy is of particular importance for high mountain permafrost research as they determine the ground thermal regime in the first place and therefore also the thermal and mechanical behaviour of the permafrost. The main aims of this paper are the following: (1) to present a long-term dataset of meteorological measurements at three different mountain permafrost sites and make this dataset available to national and international databases; (2) to quantify the energy fluxes at the surface of these permafrost sites using the long-term datasets; (3) to record and explain patterns

of variation in surface layer meteorology for the different surfaces and to compare them. These objectives are important for the development of physically based energy balance models for permafrost environments and to evaluate their sensitivity to climatic warming.

## 2 Study sites and measurements

All three investigation sites, Schilthorn, Stockhorn and Murtèl–Corvatsch are characterized by the occurrence of deep-seated

permafrost, which is currently warming at a high rate. This is shown in figure 1, where the temporal evolution of the ground temperatures in the boreholes are plotted. The measurement sites were initially instrumented for research purposes and not as long-term monitoring stations, the data have some larger gaps, which could only partly be filled. Details to the individual measurements at each location can be found in Hoelzle and Gruber (2008) and table 1. Raw measurements mainly consist of 10 minute values, which were averaged and stored as hourly means. The data is stored in Campbell loggers (CR10X or CR1000),

which are directly transferred via GSM communication to the PERMOS database.



## 2.1 Murtèl–Corvatsch

The Murtèl–Corvatsch area is situated in the Upper Engadine (Eastern Swiss Alps). A steep NW-facing rock wall surrounds the area and the slopes below the rock walls are covered by several meters of loose debris. Several rockglaciers and protalus ramparts are situated at the foot of the slope below the ridge. One of these rockglaciers is the Murtèl rockglacier, which has been investigated for more than 30 years (Hoelzle et al., 2002). It consists on a coarse blocky active layer with a thickness of about 3 - 6 m, which is situated above an ice supersaturated permafrost layer of about 25 m thickness followed by a thick layer of unersaturated blocky layer. The rocks consists on deeply weathered micaceous shales. The mean annual precipitation sum (1981-2010) measured at the nearby station of MeteoSwiss at Sils-Maria is 1011 mm whereas about 1295 mm were recorded on the Piz Corvatsch summit station. A first borehole was drilled in 1987 and has provided ground temperature data since then. This borehole was replaced by a new borehole in the year 2015 (PERMOS, 2019). In 1997, a first micrometeorological station was installed at this site (Mittaz et al., 2000). The station is composed of a 3 m high tower with various meteorological sensors, at around 5 m distance from the boreholes. Air temperature, air humidity, radiation (all components), surface temperature, snow depth, wind speed and wind direction are measured at around 2 m above ground (Hoelzle and Gruber, 2008).

## 2.2 Schilthorn

The Schilthorn is situated at the transition between the Prealps in the north and the principle chain of the Bernese Alps (Northern Swiss Alps), which exceed 4000 m a.s.l. Today, only few perennial snow patches can be found, which have been shrinking considerably as a consequence of the warm 1980s and 1990s (Imhof et al., 2000). Due to its geographic situation (exposed to moist western winds), the climate in the Schilthorn massif is of suboceanic alpine character, i.e. with fairly abundant precipitation, showing a distinct summer maximum. The mean annual sum in Lauterbrunnen (valley floor) is 1234 mm whereas about 2700 mm were estimated on the highest summits, where about 90% of the precipitation falls as snow. As a consequence, the surface is snow-covered for longer periods than in most of the comparable regions of the central Alps. The geology consists of deeply weathered sandstones, which are covered by fine grained debris of sandy and silty material with a low ice content at the Northern slope of Schilthorn summit (Imhof et al., 2000). A microclimatological station was installed on a small plateau in the north facing slope about 100 m to the NW of the Schilthorn summit at 2900 m a.s.l. in 1998. Air temperature, air humidity, radiation (all components), snow depth, wind speed and wind direction are measured at around 2 m above ground (Hoelzle and Gruber, 2008). In the same year, a 14-m deep borehole was drilled into bedrock a few meters to the northwest of the climate station and equipped with a thermistor chain. Two additional boreholes were drilled in the year 2000 with a vertical one having a depth of 100 m and one perpendicular to the surface with a depth of 92 m. A 20m deep replacement borehole was drilled in 2018 at a few meters distance to the boreholes drilled in 2000.In addition, soil moisture data are recorded nearby (Pellet and Hauck, 2017).



## 2.3 Stockhorn

This study site is located in the upper Mattertal/Walliser Alpen in the Southern Swiss Alps. The station is located on a mountain ridge which extends from the Gornergrat to the Stockhorn mountain summit. The station is situated at an altitude of 3410 m a.s.l. on a small plateau which is gently inclined to the south (45°59'N, 7°49'E) showing intermediate ice contents and heterogeneous surface conditions with medium-size debris, fine grained material and outcropping bedrock. Ice content estimation and general ground characterisation are based on geophysical surveying and borehole drilling. The bedrock belongs to the palaeozoic crystalline Monte Rosa nappe and consists on Albit-Muskovit schists and shows at some places the development of patterned ground, especially at the site where the station is located. The Stockhorn–Gornergrat crest is surrounded by mountain ranges that exceed altitudes of 4000 m a.s.l. This causes a local climate characterized by reduced cloud cover and high solar radiation. The annual precipitation in the Zermatt valley is relatively low with 639 mm according to the MeteoSwiss fro the period 1981 – 2010 and can be estimated to be around 1500 mm at Stockhorn based on King (1990). The two boreholes were drilled in July 2000. One borehole is 100 m deep and is located on the plateau, whereas a second borehole is 31 m deep and situated around 25 m to the south close to the edge of the plateau. Borehole data recording started in June 2001 (Gruber et al., 2004). A meteorological station was installed in June 2002 close to the deep borehole. Air temperature, air humidity, radiation (all components), surface temperature, snow depth, wind speed and wind direction are measured at around 2 m above ground (Hoelzle and Gruber, 2008). In addition, soil moisture data are recorded nearby (Pellet and Hauck, 2017).

## 3 Methods

### 3.1 Calculation of the Surface Energy Balance

The energy balance at the surface can be formulated as follows (energy fluxes are in units of $Wm^{-2}$ and positive when directed towards the surface):

$$Q_S{\downarrow} + Q_S{\uparrow} + Q_L{\downarrow} + Q_L{\uparrow} + Q_H + Q_{LE} + Q_S + Q_P + Q_G + Q_M = 0 \tag{1}$$

where $Q_S{\downarrow}$, $Q_S{\uparrow}$ are the shortwave incoming and reflected radiation, $Q_L{\downarrow}$ and $Q_L{\uparrow}$ are the longwave incoming and emitted radiation fluxes, $Q_H$ is the turbulent sensible heat flux, $Q_{LE}$ is the turbulent latent heat flux, $Q_S$ is the heat flux through the snow cover, $Q_P$ is the sensible heat energy supplied or consumed by precipitation falling on the surface. It is ignored as it is in general negligibly small compared with the other fluxes (Brock et al., 2010). $Q_G$ is the heat flux in the ground and $Q_M$ denotes snow melt at the surface.

### 3.1.1 Short- and longwave radiation fluxes

$Q_S{\downarrow}$, $Q_S{\uparrow}$, $Q_L{\downarrow}$, $Q_L{\uparrow}$ were measured directly at the individual meteostations using a CNR1-device at each of the three sites (for further details see table 1). The sensors were relatively reliably during the measurement period and only a few longer gaps





occurred. The sum of all individual radiation components is called the net radiation $Q_R$, which is a fundamental factor for all energy balance studies as it typically is the component with the highest energy gain or loss.

### 3.1.2 Turbulent sensible and latent heat flux

The turbulent fluxes were calculated using two methods a) the bulk aerodynamic method (e.g. Munro, 1989; Denby and Greuell,
2000; Arck and Scherrer, 2001; Oke, 1987) and b) the Bowen ratio method (Bowen, 1926).

### 3.1.3 Bulk method

Knowing that turbulent sensible and latent heat fluxes over rock and debris surfaces cannot be estimated using neutral stability assumptions (Nakawo and Young, 1982; Mattson and Gardner, 1989; Takeuchi et al., 2000; Nicholson and Benn, 2006), it is necessary to apply corrections for non-neutral conditions due to the large variations in atmospheric stability. The stability of
the atmospheric layer close to the surface can be described by the bulk Richardson number, $Ri_b$, relating the relative effects of buoyancy to mechanical forces (Brutsaert, 1982; Moore, 1983):

$$Ri_b = \frac{g(T_a - T_s)(z - z_{0m})}{T_0 u^2} \tag{2}$$

where $g$ is the acceleration due to gravity ($9.81 ms^{-2}$); $T_0$ is the mean absolute air temperature between the surface and the measurement level $z$; $z_{0m}$ is the surface roughness length for momentum taken from Brock et al. (2006, 2010) as 0.016 and
0.001 for snow free and snow covered conditions, respectively; $T_a$ is the air temperature; $T_s$ is the surface temperature and $u$ is the wind speed measured at the station. Stability corrections based on $Ri_b$ have been applied successfully over rockglaciers (Mittaz et al., 2000). Assuming that the local gradients of mean $u$, mean $T$ and mean specific humidity $q$ are equal to the finite differences between the measurement level and the surface, the turbulent fluxes may be evaluated as follows (after Brutsaert (1982); Favier et al. (2004); Sicart et al. (2005)):

$$Q_H = \rho_a \frac{c_p k^2 u (T_a - T_s)}{(ln \frac{z}{z_{0m}})(ln \frac{z}{z_{0t}})}(\phi_m \phi_h)^{-1} \tag{3}$$

$$Q_{LE} = \rho_a \frac{L_v k^2 u (q_a - q_s)}{(ln \frac{z}{z_{0m}})(ln \frac{z}{z_{0q}})}(\phi_m \phi_v)^{-1} \tag{4}$$

where $q_a$ and $q_s$ are specific humidities ($kg kg^{-1}$) at the 2 m and surface levels, respectively; $\rho_a$ is the air density; $c_p$ is the specific heat capacity for air at constant pressure ($c_p = c_{pd}(1 + 0.84q)$ with $c_{pd} = 1005 J kg^{-1} K^{-1}$); $k$ is von Karmans constant ($k = 0.4$) and $L_v$ is the latent heat of vaporisation ($L_v = 2.476 * 10^6 J kg^{-1}$ at 283 K). The scalar lengths for heat $z_{0t}$
and humidity $z_{0q}$ were considered to be equal to $z_{0m}$. $\phi_m$, $\phi_h$, $\phi_v$ are the non-dimensional stability functions for momentum, heat and moisture, respectively and they are unity in neutral cases. They are expressed as functions of $Ri_b$ for stable and unstable cases:





Stable case ($0 < Ri_b < 0.2$):

$$(\phi_m \phi_h)^{-1} = (\phi_m \phi_v)^{-1} = (1 - 5Ri_b)^2 \tag{5}$$

Unstable case ($Ri_b < 0$):

$$(\phi_m \phi_h)^{-1} = (\phi_m \phi_v)^{-1} = (1 - 16Ri_b)^{0.75} \tag{6}$$

### 5   3.1.4   Bowen ratio method

The Bowen ratio method estimates the sensible and latent heat fluxes $Q_H$ and $Q_{LE}$ according to the energy available (e.g. $Q_R + Q_G + Q_M + Q_S$). The energy partitioning between $Q_H$ and $Q_{LE}$ has direct relation to the boundary-layer climate and is called Bowen ratio $\beta$ (Bowen, 1926):

$$\beta = \frac{Q_H}{Q_{LE}} = \frac{c_p(T_a - T_s)}{L_v(q_a - q_s)} \tag{7}$$

with

$$Q_H = \frac{Q_R + Q_S + Q_G + Q_M}{1 + \frac{1}{\beta}} \tag{8}$$

and

$$Q_{LE} = \frac{Q_R + Q_S + Q_G + Q_M}{1 + \beta} \tag{9}$$

Using the Bowen ratio energy-balance method leads to some uncertainties, which has to be taken into account when inter-
preting the data. First, errors of net radiation and subsurface fluxes are accumulated in the evaluation of the turbulent fluxes. Secondly, the Bowen ratio method often produces non-physical sensible and latent heat fluxes: wrong signs (directions) and extremely large magnitudes of the fluxes. Therefore, we follow a procedure proposed by Ohmura (1982) using two criteria for rejecting such undesirable data. The first criteria is defined by the condition that data is excluded, if the conditions below are not met:

if $Q_R + Q_G > 0$ then $\Delta T > (-L_v/c_p)\Delta q$

if $Q_R + Q_G < 0$ then $\Delta T < (-L_v/c_p)\Delta q$

The second criteria is defined by the necessary condition that the denominator in the equations and cannot be zero. This case occurs when $\beta = -1$. In order to consider the accuracy of temperature and specific humidity, this condition is formulated as follows:

$\Delta T > -(L_v/c_p)\Delta q - 2[(L_v/c_p)E(q) + E(T)]$

$\Delta T < -(L_v/c_p)\Delta q + 2[(L_v/c_p)E(q) + E(T)]$

where $E_T$ and $E_q$ are the resolution limits of the thermometer or the hygrometer.



### 3.1.5 Snow heat flux

The snow heat flux is given by

$$Q_S = -k_{snow}\frac{dT}{dz} \tag{10}$$

where, according to (Keller, 1994; Mellor, 1977), the heat conductivity of snow ($Wm^{-1}K^{-1}$) is determined by $k_{snow} =$
$2.93(\rho_{snow}^2 10^{-6} + 0.01)$. Snow density $\rho_{nows}$, which was fixed to a value of $220\,kgm^{-3}$. Typical values for $k_{snow}$ lie between
0.2 and 0.5 ($Wm^{-1}K^{-1}$) (e.g. Sturm et al., 1997; Essery et al., 2013). Due to these low values, the transport of heat through
the snow cover is generally small. Hence, snow insulates the ground from atmospheric influences.

### 3.1.6 Ground heat flux

While the mean annual ground surface temperature is a function of the varying energy-exchange processes at the atmo-
sphere/lithosphere boundary, their propagation mainly downwards depends on the thermal properties of the ground (Williams
and Smith, 1989). Heat conduction is the main process of heat transfer within the ground when considering the ground as a
homogeneous medium. In addition, heat transport takes place by circulation of water and convective and advective circulation
of air. However, these processes are not included in the calculations in order to keep the approach as simple as possible:

$$Q_G = -k(\Delta T / \Delta z) \tag{11}$$

where $Q_G$ = ground heat flux and $k$ = thermal conductivity of the ground in $Wm^{-1}K^{-1}$.

### 3.1.7 Snow melt

The energy provided by snow melt is

$$Q_M = L_M R_S \tag{12}$$

where $L_M$ = latent heat of fusion for ice ($336kJkg^{-1}$). The runoff rate $R_S$ can be obtained as a function of the change of the
product of snow height $h$ and snow density $\rho_s$, which was fixed to a value of $220\,kgm^{-3}$:

$$R_S = \frac{\Delta(\rho_{snow}h)}{\Delta t} \tag{13}$$

## 3.2 Data processing

As the monitoring evolved slowly from individual research projects (e.g. PACE) into the long-term PERMOS monitoring, the
data availability at the beginning of the observation period is relatively poor. At all three stations, major data gaps up to several
months exists for all measured variables. For air temperature and snow height, being the most important meteorological vari-
ables for many permafrost modelling studies, an almost complete time series was generated by gap filling using complementary





information from nearby meteorological stations (mainly based on Intercantonal Measurement and Information System (IMIS) or MeteoSwiss (MCH) stations). For the energy balance, only shorter gaps were filled using correlations between the individual variables at the stations itself to guarantee that the energy balance is based on in-situ data gathered at the meteorological stations alone.

Several data sets are prepared for this study, which are classified into three levels. Level 0 is the collected set of raw data, mainly obtained directly from the loggers (wherever the data was still available). This data is not treated at all and large part of this data set may contain errors. Level 1 data consist of corrected and partly gap-filled data. The gap filling process is described below. Finally, level 2 data is the final data set, including additional data processing steps such as correction of shortwave incoming radiation due to snow covered sensors. For further studies, the use of the level 2 data set is recommended. The raw

data (level 0) were processed using the following processing steps in that order:

- range check and small corrections (level 1): For all stations, the data of the variable **air temperature** outside the range of –40 to +30 °C and **relative humidity** data outside the range of $0 - 101$ % were deleted. Data between 100 and 100.9 % were corrected to 100 %. **Snow height** data was corrected after comparison with neighbouring operational stations in the close surroundings of Murtèl-Corvatsch, Schilthorn and Stockhorn. Individual outliers were either deleted or corrected

manually with the help of additional information from our station measurements such as albedo. Negative incoming and outgoing **shortwave wave radiation** during night were set to 0. Unrealistic high values (> than top of atmosphere radiation) were corrected to a value corresponding to its neighbouring values. For the level 1 data, no correction was made for snow coverage on the sensor.

- short gap filling (level 1): In general, single missing values for up to three hours were corrected by taking the preceding

value to fill the unknown values. By this, air temperature and snow height data were almost completely gap-filled. In addition, site specific processing steps were performed for the different variables.

- long gap filling (level 1): For the individual stations containing data gaps larger than 3 hours, missing values were replaced using linear regression between data from nearby stations or the station itself. The corresponding variables and coefficients of the linear regressions can be found in table 2. **Snow height** data at the Murtél-Corvatsch station was

corrected by taking the snow height data from the station Lagrev IMIS with a multiplication factor of 0.625. In the case of Schilthorn the station Türliboden IMIS with a factor of 0.58016 and for Stockhorn the station Gornergrat IMIS with a multiplier of 0.369425 were used.

- radiation corrections (level 2): Level 2 data were corrected using $\alpha$ for fresh snow ($\alpha = 0.87$, (Oke, 1987)) and multiplying it with outgoing radiation. **Longwave incoming radiation** was additionally corrected for the known influence of the

shortwave incoming radiation by using the formula given by Sicart et al. (2005) $Q_L\downarrow = Q_L\downarrow$ - (corr * $Q_S\downarrow$), where corr is equal to 2%. We considered also that the radiation input perpendicular to a slope is decisive for the energy balance of a surface. The values obtained for the different radiation components have, therefore, to be adapted to the local slope. As





a consequence, the values at Murtél-Corvatsch (local slope is 2°) and at Schilthorn (local slope is 7°) were corrected by dividing the obtained measurement values with the cosine of this angle.

– data aggregation: Data were aggregated as follows: First, the data were corrected and stored as level 1 hourly data. Then the radiation corrections mentioned above were applied to produce level 2 data. From the level 2 data, the energy balance was calculated. Following a recommendation by the WMO (2017), mean daily values were then calculated from the hourly values if at least 19 hours of data were available for a full day. From the daily values, we calculated monthly mean values using a threshold of 25 days meaning that monthly means are only calculated when at least 25 days of a month were available.

A special case is the Stockhorn station, where meteorological data between 2002 and 2006 was only stored as six hourly mean values, with frequent data gaps. Therefore, subsequent energy balance calculations were only performed for the period 2006-2018, after the station was reprogrammed to record hourly values. Level 2 data could only be produced for this time period. Nevertheless, level 1 daily meteorological data for Stockhorn were produced for the whole measurement period since 2002, albeit with reduced data quality.

## 4 Results

We will provide some of the long-term time series, which are of relevance for the permafrost monitoring such as the air temperatures, snow heights or also information about the dates of the first larger snow fall and snow melt out. We will mainly present monthly aggregated values of all meteorological variables as well as of the individual energy balance components.

### 4.1 Measured meteorological components at the three sites

Figure 2 shows the evolution of the mean daily values of air temperature, snow depth and ground temperatures at all three sites. The observed mean air temperature for the individual observation periods are at Murtèl-Corvatsch (1997—2018) -1.66 °C, at Schilthorn (1999—2018) -2.60 °C and at Stockhorn (2003–2018) -6.18 °C. During the observation periods the measured air temperatures show hereby a warming of 0.7 °C at Murtèl-Corvatsch, 1.0 °C at Schilthorn and 0.6 °C at Stockhorn. Measured mean snow heights (figure 2) for the periods in winter results in values of 0.50 m at Murtèl-Corvatsch, 0.87 m at Schilthorn and 0.32 m at Stockhorn. This means that snow height at Schilthorn is close to three times the value of Stockhorn. Maximum snow height for Murtèl-Corvatsch is 2.27 m, for Schilthorn 4.01 m and for Stockhorn 2.11 m. A slightly increasing trend of snow height during the observation periods is recorded at the Murtèl-Corvatsch and Stockhorn sites and a decreasing trend is observed at the Schilthorn site. However, the trend would be also positive when the first winter in 1999 with a record snow depth would be neglected. The dates when the first snow cover starts to build up in autumn and when the snow melt is completed in spring was determined from the snow height sensor resulting in the amount of snow free days. Figure 3 illustrates that the number of snow free days during the observation periods have a clear increasing tendency. The evolution of the near surface ground temperatures measured within the boreholes is shown in figure 2. The mean ground temperatures are slightly negative





for Stockhorn with -0.43 °C at a depth of 0.3 m and -0.32 °C at 0.8 m, but positive for Schilthorn (0.03 °C at a depth of 0.2 m and 0.04 °C at 0.4 m) and Murtèl-Corvatsch (0.07 at a depth of 0.5 m and -0.28 °C at 1.5 m). This reveals that the observed permafrost at lower depths at these places is either not stable anymore on the longer term or that other processes in the active layer are producing a considerable offset, as it is the case at the Murtèl-Corvatsch site with its very coarse blocky

surface (Hanson and Hoelzle, 2004; Scherler et al., 2014).

The shortwave incoming and outgoing radiation show mean values of 147.91 $Wm^{-2}$ and -71.50 $Wm^{-2}$ for Murtèl-Corvatsch, 149.61 $Wm^{-2}$ and -75.87 $Wm^{-2}$ for Schilthorn and 209.69 $Wm^{-2}$ and -131.32 $Wm^{-2}$ for Stockhorn, respectively, resulting in mean albedo values of 0.48 for Murtèl-Corvatsch, 0.51 for Schilthorn and 0.63 for Stockhorn. For the longwave incoming and outgoing radiation the following values were measured, 254.40 $Wm^{-2}$ and -300.38 $Wm^{-2}$ for

Murtèl-Corvatsch, 254.97 $Wm^{-2}$ and -296.02 $Wm^{-2}$ for Schilthorn and 213.35 $Wm^{-2}$ and -284.78 $Wm^{-2}$ for Stockhorn, respectively (see figure 4 and table 3). The corresponding net shortwave and longwave balance are 76.41 $Wm^{-2}$ and -45.41 $Wm^{-2}$ for Murtèl-Corvatsch, 73.34 $Wm^{-2}$ and -41.05 $Wm^{-2}$ for Schilthorn and 78.37 $Wm^{-2}$ and -71.43 $Wm^{-2}$ for Stockhorn, respectively (see figure 4 and table 3). At all sites the radiation components show a slight decreasing trend of incoming shortwave radiation and an increasing trend for incoming longwave radiation - except for the Schilthorn site. At all

sites the mean observed relative humidity is close to 70% and shows a clear seasonal behaviour with monthly values close to 60% during the winter and values of around 80% during summer. Considering wind speed and wind direction we observe that at Murtèl-Corvatsch the wind speed has a mean value of 1.69 $ms^{-1}$, which is much lower than at Schilthorn with a mean wind speed of 1.97 $ms^{-1}$ or Stockhorn with 2.09 $ms^{-1}$ (see table 3). Wind direction at Murtèl-Corvatsch and Schilthorn are particularly influenced by topography as both stations are located on Northern exposition and are influenced by mountain crests to

the South. The Stockhorn station is situated on a mountain crest and a predominant wind direction from East-Southeast (figure 5).

## 4.2   Energy Balance

The energy balance at the surface is particularly important as it defines the boundary conditions for the energy transfer between the atmosphere and subsurface. It has major impacts on the ground heat flux and therefore also on the occurrence of permafrost.

The components of the energy balance at Stockhorn are smaller than at the other two sites (figure 6). Net radiation is only about 20% of the net radiation at Schilthorn and Murtèl-Corvatsch. The share of the energy balance components is of particular interest. At all three sites the most important energy balance component is the net radiation, which has a net positive value (see figure 6). However, the monthly values show strong seasonal fluctuations (figure 7). At Stockhorn, the net radiation is negative from October to April whereas at Schilthorn and Murtèl-Corvatsch, it is only negative between November and February (figure

7). The influence of the snow cover on the energy fluxes is also of importance. Particularly in spring, snow is impacting the incoming shortwave radiation by its high albedo and we can recognize a strong difference between the sites. Schilthorn generally has a thick snow cover and in the months of April to July much of the available atmospheric energy is used for its melting (mean value of 8.46 $Wm^{-2}$, figure 6, table 4), which is much more than at the other two sites with a mean value of 2.26 $Wm^{-2}$ at Stockhorn and 4.17 $Wm^{-2}$ at Murtél-Corvatsch (figure 6, table 4).





The results for the three sites show that the turbulent fluxes vary between 6 to 15 $Wm^{-2}$ using the Bowen ratio method and between 3 to 15 $Wm^{-2}$ using the bulk method (table 4). The results of both methods correlate relatively well with coefficients of determination ($r^{-2}$) between 0.5 and 0.7. In comparison to the other fluxes, the heat fluxes in the snow and in the ground are very small with mean values lower than 1 $Wm^{-2}$ (figure 6, table 4).

## 5    Discussion

All three stations show a clear atmospheric warming trend. However, the warming is different at all stations. At Schilthorn (period 1999 – 2018) and Stockhorn (period 2003 – 2018), we see the strongest increase in air temperature with around $0.053\,^{\circ}Cyr^{-1}$ and $0.054\,^{\circ}Cyr^{-1}$, respectively whereas Murtèl-Corvatsch (period 1997 – 2018) experiences the smallest air temperature increase with $0.034\,^{\circ}Cyr^{-1}$. However, the temperature increase is different when only considering a period where

data is available from all three stations. Taking into account only the observation period 2003 – 2018, the trend changes particularly at the Murtèl-Corvatsch site with an increase of $0.073\,^{\circ}Cyr^{-1}$. For the same period, Schilthorn recorded a temperature increase of $0.064\,^{\circ}Cyr^{-1}$. The temperature rise is in general slightly higher when compared to the observed decadal increases between 1991 – 2020 determined regionally by (MeteoSwiss), but our observations are only covering the last two decades, where the atmospheric warming was most pronounced.

An interesting fact is a decrease in shortwave incoming radiation observed at all sites, most pronounced at Schilthorn. We cannot explicitly explain this observation, but it might well be related to an increase in cloudiness. The melt energy used during the spring season at all three sites seems to be of great importance. The large temporal variability of the snow cover at the individual sites and the strong spatial differences between them are significant. Schilthorn shows the thickest and long-lasting snow cover during the observation period and its isolating effect and energy sink during melt is probably key to keeping the

permafrost conditions active at this site. If the number of snow days in spring and early summer decreases (as projected under future climate scenarios, see (Scherler et al., 2013; Marmy et al., 2016), then the absorbed heat at the surface will increase with a corresponding increase of ground heat flux downward. We can estimate this potential increase if we compare the observed July (11.51 $Wm^{-2}$) or August (8.32 $Wm^{-2}$) ground heat flux with the ground heat flux of May (0.98 $Wm^{-2}$) and June (4.28 $Wm^{-2}$). These estimates are partly confirmed by the observed increase of $Q_G$ downwards into the ground, particularly

at Murtèl-Corvatsch and Schilthorn for June and at Stockhorn for May (figure 8). An increasing trend of snow free days has already been observed and is pointing also to a corresponding change of the heat fluxes within the last two decades (see figure 3).

Furthermore, clear differences in the radiation components at the three sites can be seen. Stockhorn is situated at higher elevation and the recorded observations show lower air temperatures as well as correspondingly lower incoming longwave

radiation (figure 6). However, shortwave incoming radiation is much higher in contrast to the other two sites leading to a higher relative importance of the shortwave incoming radiation on the energy balance at Stockhorn (figure 6). At higher altitudes, in general higher amounts of solar radiation and higher insolation variability are observed than at lower altitudes. In addition, Stockhorn is a relatively dry location with lower humidity and correspondingly lower cloudiness. These observations are of





particular interest for the question of elevation dependent warming (EDW) (Wang et al., 2016; Pepin et al., 2015). As stated in Pepin et al. (2015), a combination of mechanisms are probably responsible for the more pronounced warming trends observed at high altitude stations: (i) the albedo feedback due to retreating snowlines (e.g. Scherrer et al., 2012), (ii) the cloud feedback caused by increased latent heat release (Ohmura, 2012), and (iii) the water vapour-radiative feedbacks, which is impacting the

relationships between longwave radiation, moisture and thermal regimes and are particularly enhanced at low temperatures where already small water vapour increases can lead to a substantial influence on the incoming longwave radiation (Pepin et al., 2015). In addition to the feedback mechanisms mentioned above, the relationship between temperature and outgoing longwave radiation is a consequence of the blackbody emission, which is proportional to the fourth power of temperature (Stefan–Boltzmann law). Therefore, if the outgoing longwave radiation is increasing, a larger temperature change at lower

temperatures will result.

The albedo feedback is observed at all three sites as the number of days without snow is strongly increasing. Regarding the cloud feedback, we find evidence for this effect at all stations by the above mentioned observation of reduced shortwave incoming radiation and increased temperatures. This may be the most important factor for the strong warming observed at the Schilthorn site. The water vapour-radiative feedbacks may also explain the observed difference between the comparatively dry

Stockhorn and the other sites related to the stronger ground temperature increase. However, the overall resulting response at the individual sites is complex and the above mentioned effects will impact the different sites with different magnitude. For a better understanding, the presented data from the three stations should be combined with data from other stations in the Alps to gain a better understanding of the effects of EDW.

A significant result of this study is depicting the high dependency of the ground heat flux on the net radiation (see figure

9). It is especially interesting that a single linear relation from net radiation can be used to calculate the ground heat flux at all three stations with reasonable accuracy. We want to point to the fact that it could be of special interest for the development of simplified permafrost distribution models based on easy to calculate input parameters. In this case a rough land classification in combination with the calculated radiation fluxes would allow to use the relation shown in figure 9. It could also be shown that the air temperature is another important factor for the ground heat flux, which is clearly visible in figure 10. However, it

has to be noted that the linear relations show different slopes which does not allow for a universal application to a larger region as the individual relations at each station have to be taken into account.

Finally, thermal offsets relate the air temperatures to the ground temperatures, which is of relevance in most atmosphere–permafrost investigations. In this study, we observe a very large offset at Stockhorn when compared to the other two sites (see figure 11). In general, the thermal offset is strongly influenced by the snow cover and by the amount of incoming shortwave

and longwave radiation. The meteorological observations at the three sites revealed that the Stockhorn site receives more incoming shortwave than longwave radiation in comparison to Schilthorn and Murtél-Corvatsch. As shortwave radiation has a pronounced impact on the surface temperature and subsequently also the ground temperatures while the ground is snowfree, the resulting thermal offset is larger at this site with higher shortwave radiation compared to sites where the air temperature and the corresponding incoming longwave radiation are dominant in the energy balance. This type of observations related to

the thermal offset of different sites could also be detected along a latitudinal transect from the Arctic to Alps showing a higher



thermal offset in the Alps than in the Arctic (Etzelmüller et al., 2020), which seems to be influenced by same conditions such as a thicker snow cover and stronger incoming shortwave radiation.

## 6    Conclusions

We compiled, processed and quality checked with basic criteria the long-term meteorological data sets from three high mountain stations in the Swiss Alps with permafrost occurrence, which are embedded in the Swiss Permafrost Monitoring Network. The meteorological measurements cover a period of about two decades. The collected data provide valuable information allowing an in depth insight into the surface processes at the different permafrost observation sites. This shows clearly the advantage of long-term monitoring stations in comparison to only short-term project observations where certain long-term trends and changes cannot be detected. The data is freely available for all interested researchers on different data platforms and their corresponding databases. We have quantified different energy fluxes at the surface of the three measurement sites with permafrost occurrence. The data is available on different data platforms. Our results show that there are considerable differences in the energy fluxes at the different sites. The energy balance data is particularly important and should be used when interpreting long-term climate related changes of the past as well as to reveal the drivers of past, present and future changes at the individual sites. This is important for modelling approaches as it is showing that not only one single variable such as air temperature alone should be used. Future studies can use the existing data to explore how various permafrost sites will react to the atmospheric warming.

## 7    Data availability

The data presented and described in this study is available for download at http://dx.doi.org/10.13093/permos-meteo-2021-01 (Hoelzle et al., 2021). The data will be also integrated within different databases such as PERMOS or the the Global Energy Balance Archive.

*Author contributions.*   Contribution

MH did the analysis and wrote most of the text, TM, helped with the analysis and created most of the figures, CH, JN, CP and MS helped with creating some figures and writing and interpreting the data.

*Acknowledgements.*   We want to particularly thank all PERMOS contributers, and to the project participants of the TEMPS project. The study was mainly financed by the University of Fribourg. A special thanks goes to Cathrine Stocker-Mittaz, who started these measurements at the Murtèl-Corvatsch and the Schilthorn sites during her PhD within the PACE project and to Lorenz King and Stephan Gruber who started the measurements at the Stockhorn site. A particular thank goes to Suryanarayanan Balasubramanian who helped with the programming.



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

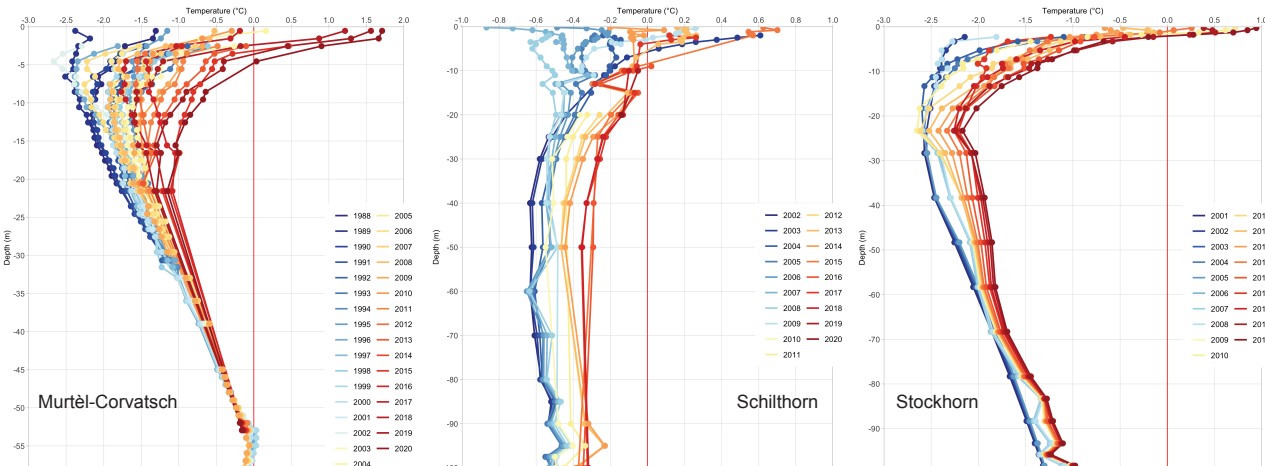

**Figure 1.** Borehole temperatures at all three stations.

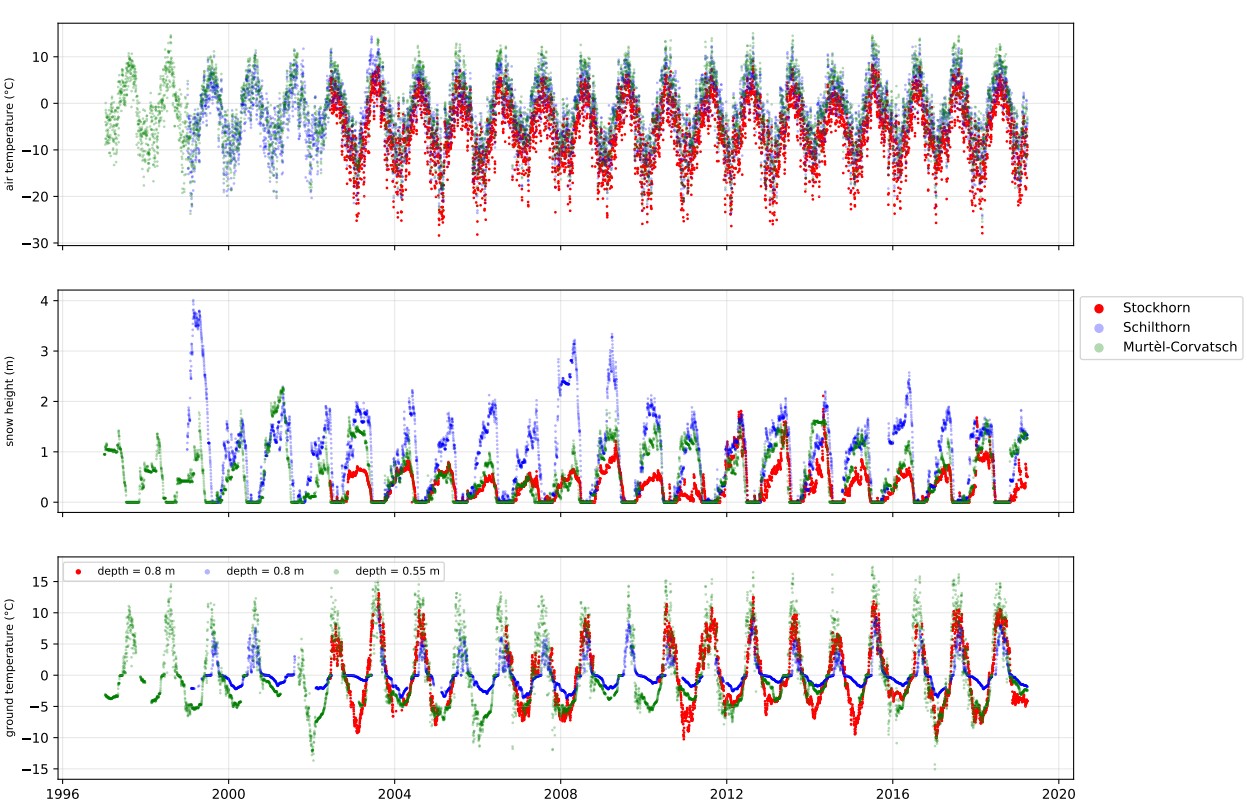

**Figure 2.** Air temperatures, snow height and ground temperatures for all three stations.



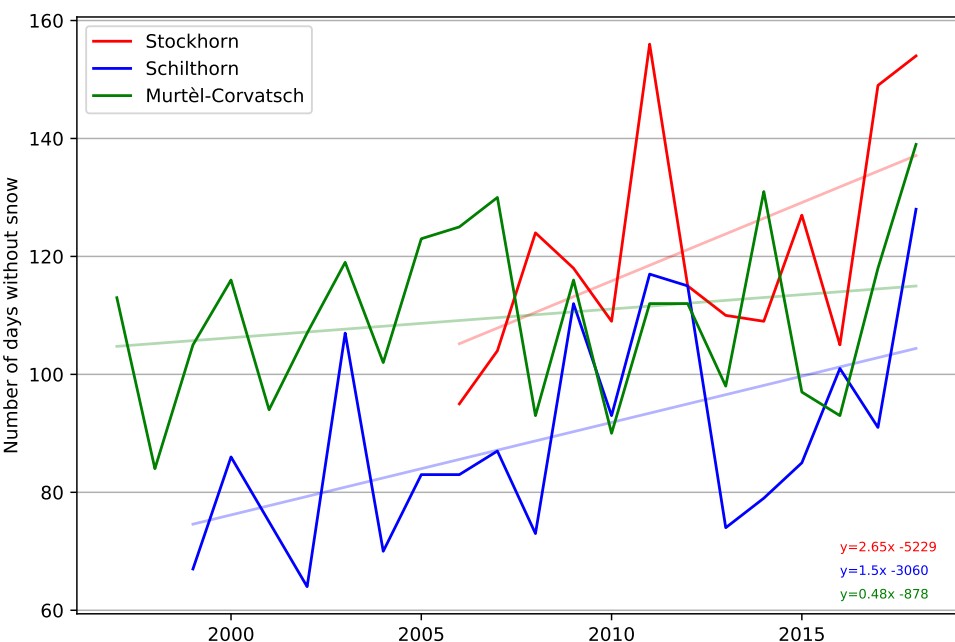

**Figure 3.** Number of days without snow for the three stations. The tendency to have longer periods without snow is clearly visible.



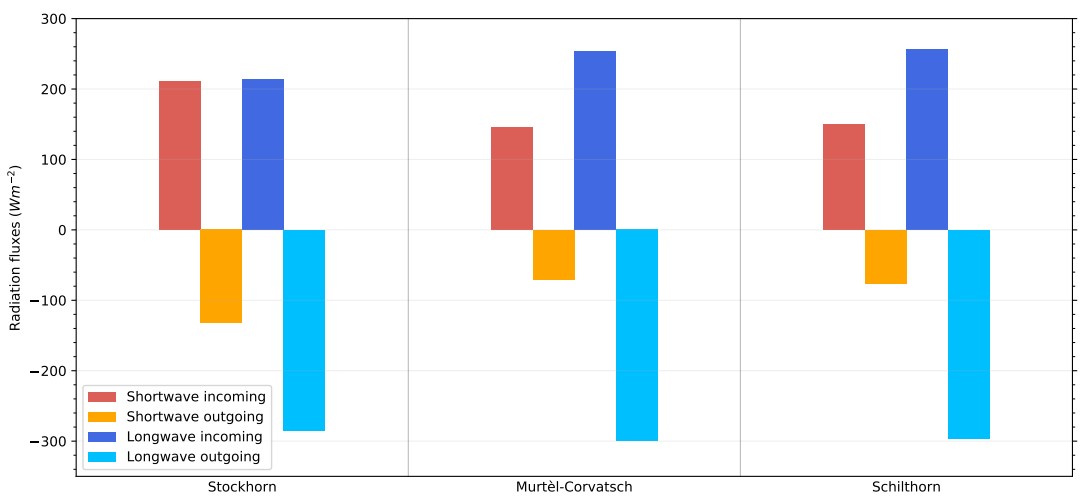

**Figure 4.** Radiation components for all three stations averaged over the observation periods (Murtèl-Corvatsch 1997 – 2018, Schilthorn (1999 – 2018, Stockhorn 2006 – 2018).



**Figure 5.** Wind speed and direction at all three stations averaged over the observation periods (Murtèl-Corvatsch 1997 – 2018, Schilthorn (1999 – 2018, Stockhorn 2006 – 2018). Note that scales of relative frequencies are different in the different subplots



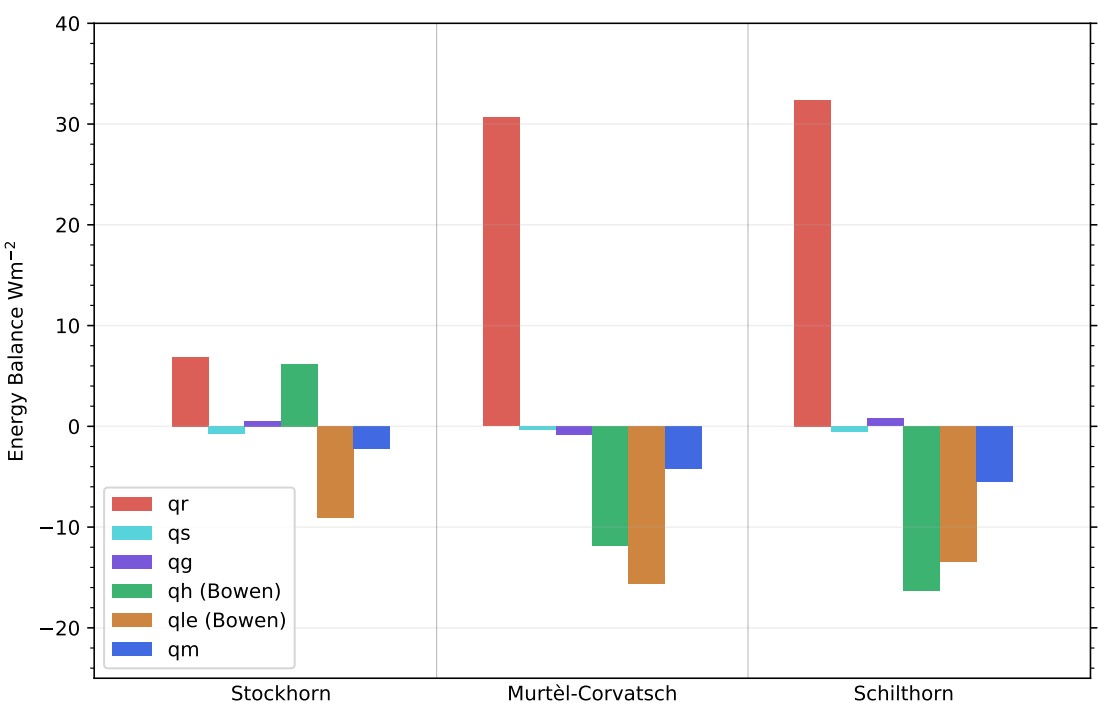

**Figure 6.** Mean energy balance components for all three stations for the whole observation period (Murtèl-Corvatsch 1997 – 2018, Schilthorn (1999 – 2018, Stockhorn 2006 – 2018).

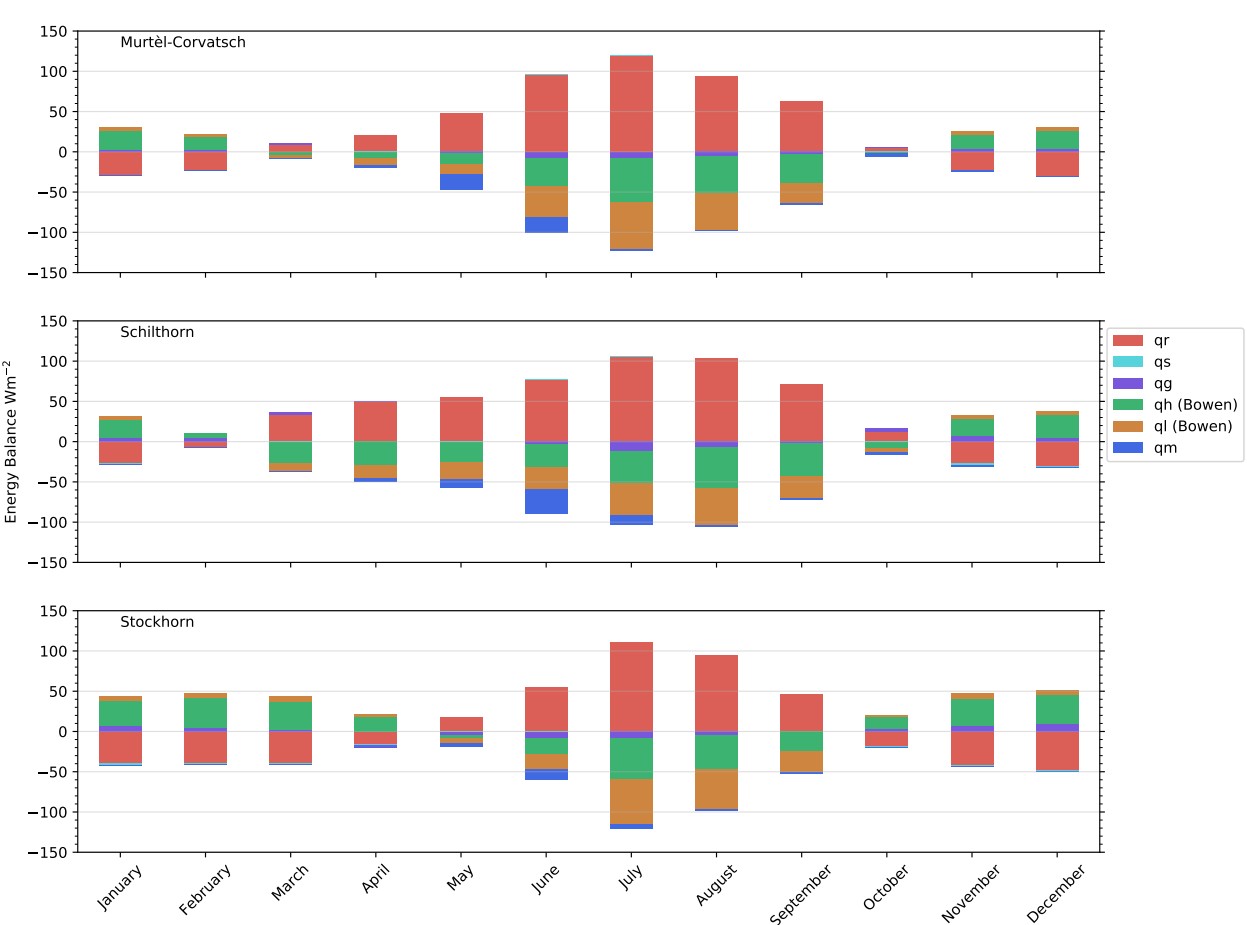

**Figure 7.** Mean monthly energy balance components for all three stations for the whole observation period (Murtèl-Corvatsch 1997 – 2018, Schilthorn (1999 – 2018, Stockhorn 2006 – 2018).



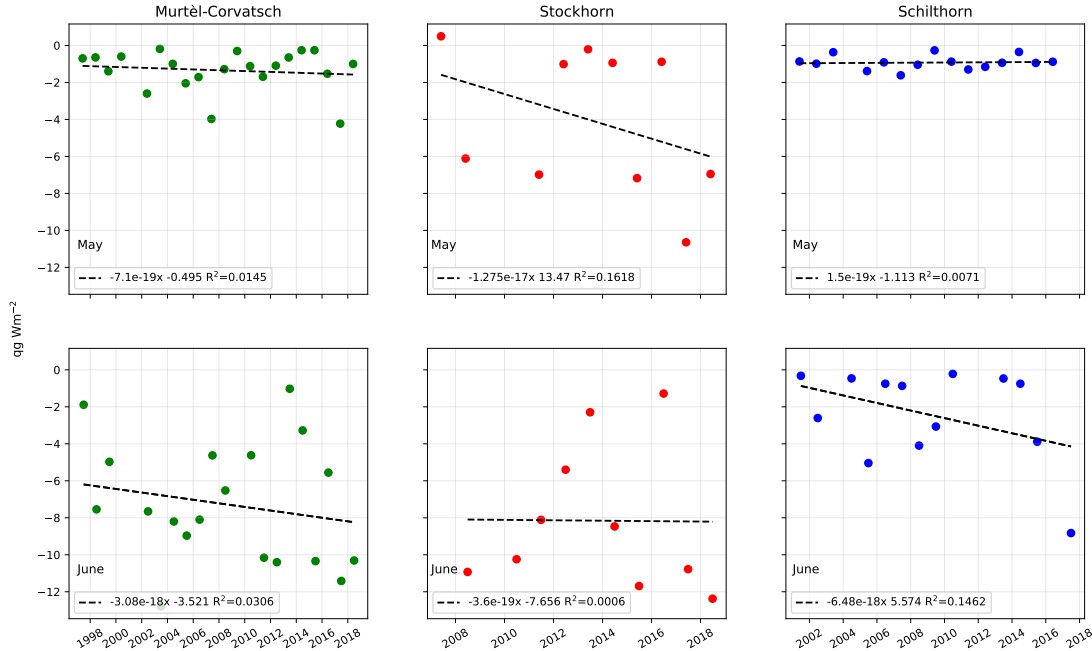

**Figure 8.** Mean ground heat fluxes for the months May and June for all observation years and for all three stations (Murtèl-Corvatsch 1997 – 2018, Schilthorn (1999 – 2018, Stockhorn 2006 – 2018)

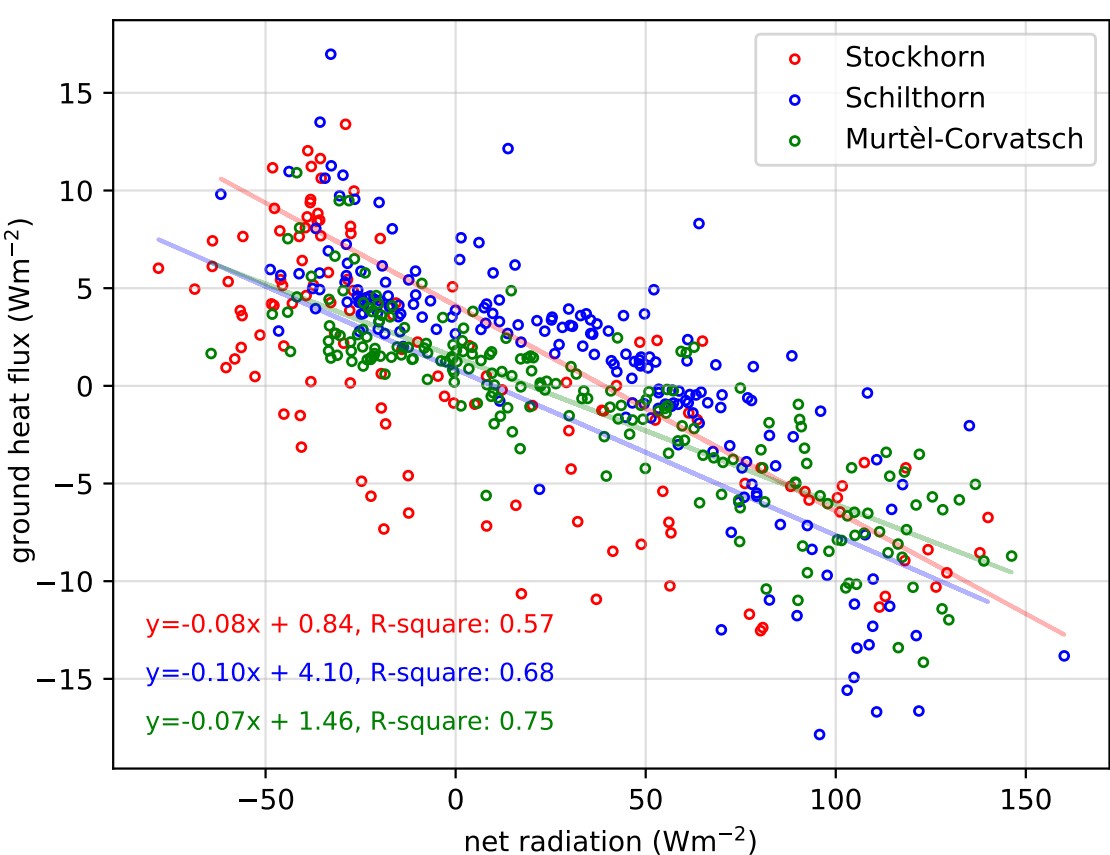

**Figure 9.** Net radiation versus ground heat flux monthly values for the whole observation periods (Murtèl-Corvatsch 1997 – 2018, Schilthorn (1999 – 2018, Stockhorn 2006 – 2018)



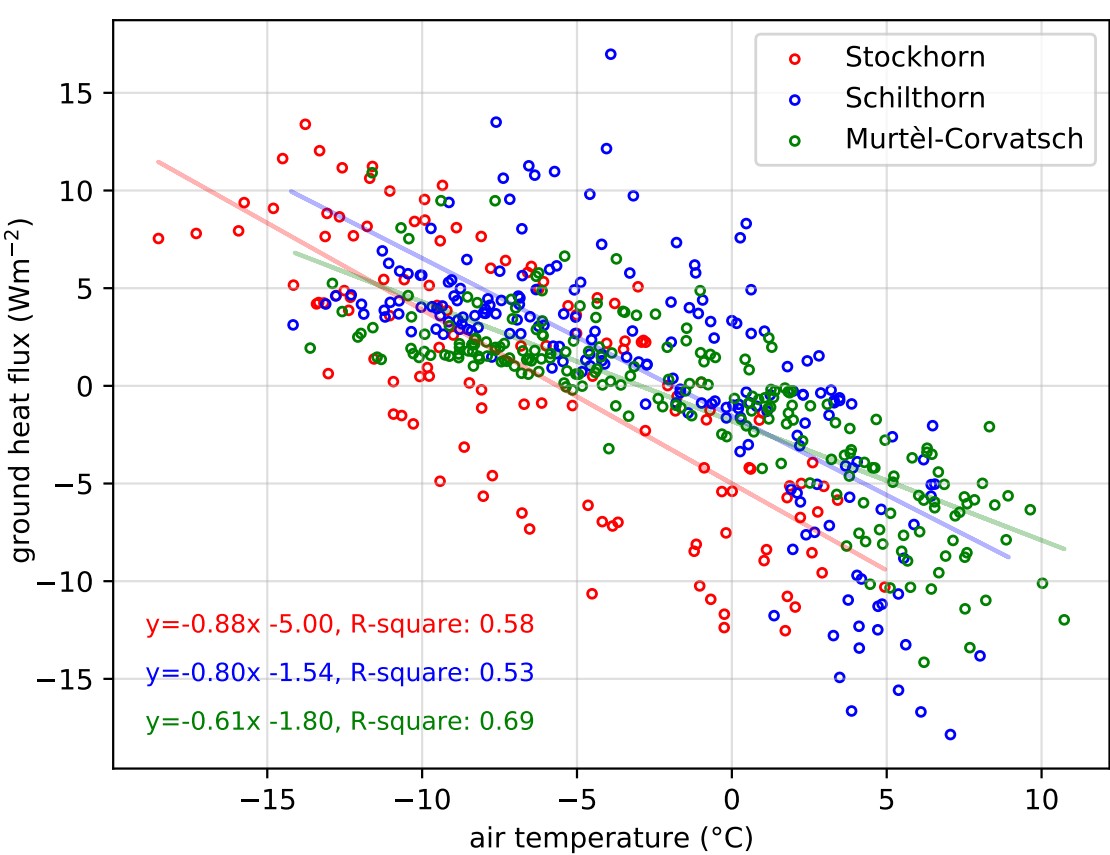

**Figure 10.** Air temperature versus ground heat flux, monthly values for the whole observation periods (Murtèl-Corvatsch 1997 – 2018, Schilthorn (1999 – 2018, Stockhorn 2006 – 2018)

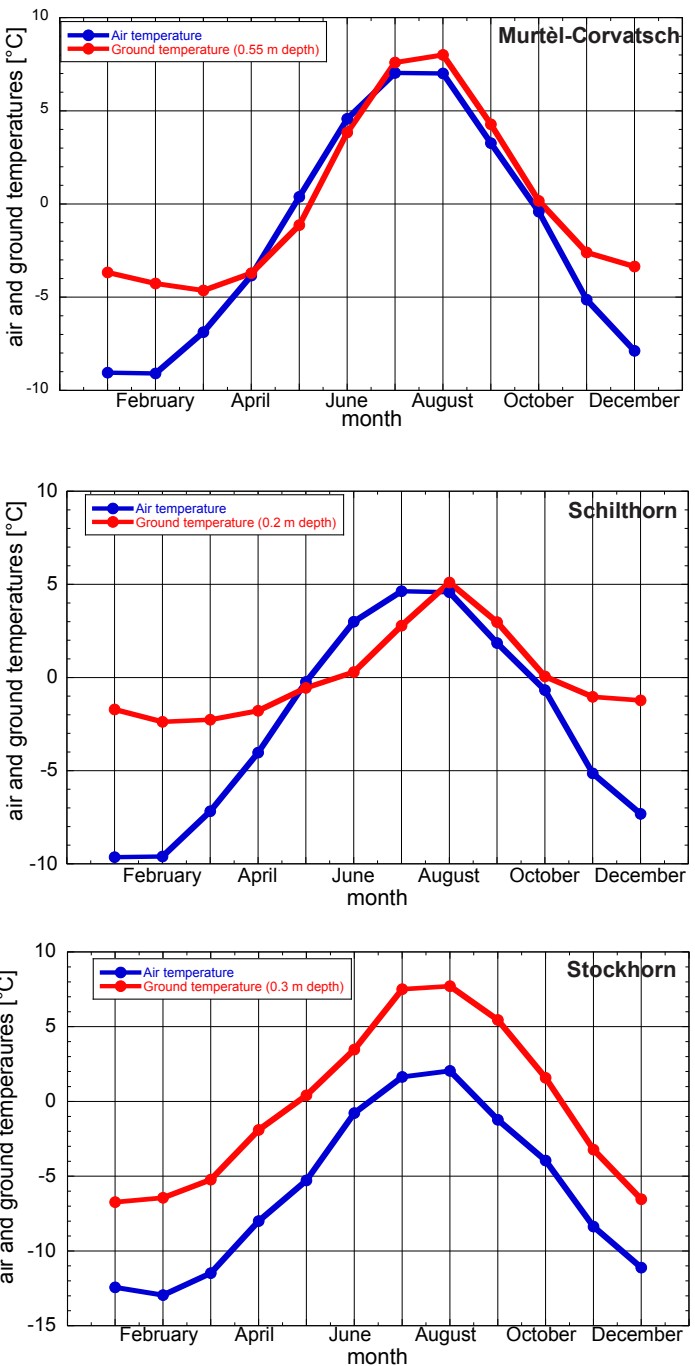

**Figure 11.** Monthly thermal offsets for all three stations for the whole observation periods (Murtèl-Corvatsch 1997 – 2018, Schilthorn (1999 – 2018, Stockhorn 2006 – 2018



**Table 1.** Sensors currently used and their characteristics at the three different sites

| variable | sensor (company) | type of sensor | range | accuracy by manufacturer | station |
|---|---|---|---|---|---|
| | logger CR10X and CR1000 (Campbell) | | | | |
| Radiation (short- and longwave) | Netradiometer CNR1 (Kipp&Zonen) | Pyranometer CM3 Pyrgeometer CG3 PT-100 T-sensor | $0.3 - 3\,\mu m$ $5 - 50\,\mu m$ | $\pm 10\%$ for daily total $\pm 10\%$ for daily total $\pm 10\%$ for daily total | Murtèl-Corvatsch Schilthorn Stockhorn |
| Air temperature Air humidity | MP-100A Ventilated (Rotronic) | RTD PT-100 C94 hygrometer | $-40 - 60\,^{\circ}C$ $0 - 100\,\%$ rh | $\pm 10\%$ $\pm 2\%$ | Murtèl-Corvatsch Schilthorn Stockhorn |
| Wind speed Wind direction | Model 05103 – 5 (Young) | Potentiometer | $0 - 60\,ms^{-1}$ $0 - 355\,^{\circ}$ | $\pm 0.3\,ms^{-1}$ $\pm 3\,^{\circ}$ | Murtèl-Corvatsch Schilthorn Stockhorn |
| Snow height | SR50 (Campbell) | Ultrasonic electrostatic transducer | $0.5 - 10$ m | $\pm 0.01\,m$ | Murtèl-Corvatsch Schilthorn Stockhorn |
| Surface temperature | Infrared thermometer | Irt/c.5 | $-35 - 60\,^{\circ}C$ | $\pm 1.5\,^{\circ}C$ | Murtèl-Corvatsch |
| Borehole temperature | YSI 44006 (Yellow Springs Instruments) UUB 31J1 (Fenwal) | NTC-thermistors | $-10 - 40\,^{\circ}C$ | $\pm 0.02\,^{\circ}C$ | Murtèl-Corvatsch Schilthorn Stockhorn |



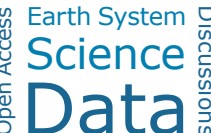

**Table 2.** Information on the linear regression coefficients used for the gap filling of level 2 data (MCH = MeteoSwiss, IMIS = Intercantonal Measurement and Information System)

Murtèl–Corvatsch

| Variable | Station used for gap filling | Coefficient $a$ | Coefficient $b$ | Correlation $r^2$ |
|---|---|---|---|---|
| Air temperature | Corvatsch MCH | 1.0322 | 3.354 | 0.95 |
| Surface temperature | Murtèl-Corvatsch Air | 1.366 | -0.896 | 0.86 |
| Relative humidity | Corvatsch MCH | 0.5817 | 24.109 | – |
| Longwave out | Murtèl-Corvatsch Air | -6.0843 | -309.98 | 0.96 |
| Longwave in | Murtèl-Corvatsch Air | 3.9944 | 260.58 | 0.78 |

Schilthorn

| Variable | Station used for gap filling | Coefficient $a$ | Coefficient $b$ | Correlation $r^2$ |
|---|---|---|---|---|
| Air temperature | Schilthorn IMIS1 | 0.98666 | -0.20132 | 0.97 |
| Relative humidity | Schilthorn IMIS1 | 0.8889 | 6.19 | 0.77 |
| Wind speed | Schilthorn IMIS1 | 0.2679 | 0.9511 | 0.28 |

Stockhorn

| Variable | Station used for gap filling | Coefficient $a$ | Coefficient $b$ | Correlation $r^2$ |
|---|---|---|---|---|
| Air temperature | Gornergrat MCH | 0.97838 | -3.847 | 0.96 |
| Ground temperature | Stockhorn ground temp. 0.3 m | -0.245 | 1.1379 | 0.92 |



**Table 3.** Mean values and standard deviation for the climate variables calculated based on the mean monthly values and the time period of measurements

|  | Mean air temperature (°C) | Mean relative humidity (%) | Mean wind speed ($ms^{-1}$) |  |
|---|---|---|---|---|
| Stockhorn | -6.18±5.63 | 73.20±5.49 | 2.09±0.57 |  |
| Schilthorn | -2.60±5.55 | 72.73±6.34 | 1.97±0.48 |  |
| Murtèl–Corvatsch | -1.66±6.12 | 71.69±4.95 | 1.69±0.32 |  |
|  | Shortwave incoming radiation ($Wm^{-2}$) | Shortwave outgoing radiation ($Wm^{-2}$) | Longwave incoming radiation ($Wm^{-2}$) | Longwave outgoing radiation ($Wm^{-2}$) |
| Stockhorn | 209.69±82.57 | -131.32±64.61 | 213.35±27.07 | -284.78±37.27 |
| Schilthorn | 149.61±88.10 | -75.87±53.73 | 254.97±26.72 | -296.02±30.05 |
| Murtèl–Corvatsch | 147.91±88.21 | -71.50±58.79 | 254.40±27.19 | -300.38±38.22 |
|  | Mean snow height (m) | Mean ground temperature (°C) | Mean ground temperature (°C) |  |
| Stockhorn | 0.32±0.31 | -0.43±5.69 (0.3m) | -0.32±4.79 (0.8m) |  |
| Schilthorn | 0.87±0.76 | 0.03±3.03 (0.2m) | 0.04±2.66 (0.4m)  height |  |
| Murtèl–Corvatsch | 0.50±0.50 | 0.07±5.15 (0.5 m) | -0.28±3.74 (1.5 m) |  |



**Table 4.** Mean values and standard deviation for the energy balance components calculated based on the mean monthly values and the time period of measurements

| | Mean net radiation balance $(Wm^{-2})$ | Mean melt energy $(Wm^{-2})$ | Mean snow heat flux $(Wm^{-2})$ | Mean ground heat flux $(Wm^{-2})$ |
|---|---|---|---|---|
| Stockhorn | 6.91±57.38 | -2.26±5.42 | -0.73±0.81 | 0.51±6.46 |
| Schilthorn | 32.40±49.02 | -8.46±16.14 | -0.52±0.65 | 0.68±6.08 |
| Murtèl–Corvatsch | 30.59±52.62 | -4.17±8.85 | -0.31±0.45 | -0.88±4.56 |
| | Mean sensible heat flux (bulk) $(Wm^{-2})$ | Mean latent heat flux (bulk) $(Wm^{-2})$ | Mean sensible heat flux (Bowen) $(Wm^{-2})$ | Mean latent heat flux (Bowen) $(Wm^{-2})$ |
| Stockhorn | -7.16±21.91 | -8.37±10.23 | 6.52±23.76 | -9.00±22.78 |
| Schilthorn | -6.80±23.63 | -6.161±12.695 | -13.70±31.25 | -13.87±17.54 |
| Murtèl–Corvatsch | -15.28±42.80 | -3.43±16.08 | -12.26±28.68 | -15.67±22.24 |