# Peer review of "Long-term energy balance measurements at three different mountain permafrost sites in the Swiss Alps"

_Earth System Science Data, 2021_

## Author Comment (AC2)

Dear Editor,

We would like to thank both reviewers for the very positive feedbacks and the constructive evaluation of the submitted article. We addressed all comments as good as possible. Particularly, we improved the figures according to the suggestions.
Below, we respond to all comments and state how we addressed them in the revised manuscript.

**Reviewer #1**

The manuscript by Hoelzle et al presents a long (20+ years) time series of energy balance measurements at alpine sites where they also measure ground temperatures. This data series is unique, at least in mountain permafrost research, and a base line for further research related to understand the influence of the different energy fluxes on ground temperatures, and thus how future climate change will influence mountain permafrost.

The manuscript is well written and documented, and I have no comments to the contents. The paper documents the data and discusses the energy balance components in relation to the observed ground temperature changes during the monitoring period, which was different at the different sites.

*We would like to thank the reviewer for these nice comments to the long-term data series of the energy balance measurements.*

Some comments and recommendations for revisions are related to the illustrations, which in general have far too small annotations, making them hardly readable. In addition, the figure text to the illustrations is very short, and one always has to rely on the main text:

*We improved the figure text to the illustrations for better readability:*

Figure 1: cannot read legend and axis labels

*We increased legend and axis label size.*

Figure 2: ditto

*We increased legend and axis label size.*

Figure 3: maybe give the significance of the trend. Maybe use only markers as the lines make the diagram very busy.

*We calculated the trend significance using the Mann-Kendall test, as suggested by Reviewer 2. We tested a plot version without the lines, however, this looked very chaotic to us. Therefore, we decided to keep the lines in the plot and hope this is ok with you.*

Figure 5: Give some larger labels. Why did you use different scale?

*We increased the label size and are now using the same scale for all subplots.*

Figure 8: small labels. Give significance of R2.

*We calculated the trend significance with the Mann-Kendall test, as suggested by Reviewer 2, and, therefore, removed $R^2$. We also increased the label size.*

Figure 9 and 10: use R^2 (not R-square). These figures has perfect label size …

*We adapted the text as suggested. We used this figures sizes to improve the label size of the other figures. Thank you for this comment.*

**Reviewer #2**

The manuscript presents meteorological and energy balance data collected at three high-altitude sites belonging to the Swiss PERMOS network. The dataset is unique and of wide interest. I really wish to thank the authors for sharing this golden dataset. The manuscript is well written and organized and almost ready for publication. I only have one major comment and few minor issues listed below.

*We would like to thank the reviewer for these constructive comments to the long-term data series of the energy balance measurements.*

In the results and discussion sections, temporal trends of many parameters (air temperature, snow height, snow cover duration, radiation, ...) are presented. I recommend the use of Mann–Kendall nonparametric test to test trends significance and Sen-slope to estimate trend values and uncertainties, rather than linear regression. I suggest including either in the text or in the figures (e.g. fig3 and fig8) trend values, uncertainty, and significance.

*Thanks very much for this very important comment, which we believe is a real improvement for the paper. We took all suggestions into account and used the Mann-Kendall nonparametric test to show the trend significance and Sen-slope to estimate trend values and uncertainties and removed the linear regression.*

Minor and technical points

I share the comment of reviewer 1 regarding figures: axis and labels size are often too small

*Thanks for this comment. See our comment for reviewer 1, how we have changed the figures according to the suggestions.*

The relevance of interplay between surface energy balance and surface cover is outlined in many sentences of the introduction (e.g. p2 l23, p3 l7, ....). I found it a bit difficult to understand Schiltorn surface characteristics. Is it "fine-grained debris of sandy and silty material" (p4 l22)? Corvatsch surface is a "coarse blocky rock glacier" (p4 l5); Stockhorn surface is "medium-size debris, fine-grained material, and outcropping bedrock" (p5 l5). Am

I right? Maybe a table summarising site characteristics (elevation, mean snow depth or duration, surface characteristics, ...) could be of help?

*We reformulated the text in the description of the sites to better include the site characteristics.*

p3 l27: " ... as long-term monitoring stations, the data have some larger gaps, which could only partly be filled." -> maybe " ... as long-term monitoring stations; the data have some larger gaps, which could only partly be filled."

*We have corrected the sentence according to the suggestion of the reviewer.*

p3 l29: Maybe "Data are stored ... CR1000) and are directly ... " could be better?

*We have corrected the sentence according to the suggestion of the reviewer.*

p4 l16: "Today, only few perennial snow patches can be found, which have been shrinking considerably as a consequence of the warm 1980s and 1990s (Imhof et al., 2000)." Is this sentence really needed?

*We removed this sentence from the manuscript.*

p4 l29: typo: "2000.In addition"

*We corrected the typo and introduced a space character.*

p5 l10: typo: "MeteoSwiss fro the"

*We corrected the typo.*

p5 l21 fla1: I find the use of the notations $Qs\downarrow$ and $Qs\uparrow$(shortwave radiation) and $Qs$ (snow heat flux) potentially misleading. I suggest using something like $Qsw$ for radiation and $Qs$ for snow to avoid the risk of confusion

*We changed the notations to the suggested form of $Qsw\downarrow$ and $Qsw\uparrow$ for the shortwave radiation to clearly distinguish it from the snow heat flux $Qs$.*

p6 l13: "$T0$ is the mean absolute air temperature between ... ". is there a word missing here? $T0$ is the mean absolute air temperature "difference" between ....?

*$T0$ is the mean value of $(Ta + Ts)/2$. We improved the formulation.*

p6 l23 fla (4) and p7 fla (7): I imagine $qa$ is calculated from relative humidity measurements at 2m. What about $qs$?

*Yes, $qa$ is calculated from the relative humidity measurements and $qs$ is adapted to the surface: if it is snow covered full saturation is assumed, if it is bare rock a strongly reduced saturation of only 10% is used.*

p8 l4-7 & l20: What is the influence of the choice of using constant snow density values (220 kgm-3) on the computation of QS and QM and thus on the partitioning of all energy fluxes (e.g. fig 7 and p12 l16-20)? Can you comment on that?

*Thanks for this very good question. We only measure the snow height at our stations and have no information about the snow density. We tried to calculate the respective energy fluxes with different snow density values. However, the selected snow density of 220 kgm^-3 worked best for the full winter/spring period.*

p8 fla (11): Did you use constant k values? which ones? Which is the â   z range used for the calculation?

*Yes, we used a constant k value based on the measurements according to the thermal conductivity samples determined at Murtèl-Corvatsch (after Vonder Mühll and Haeberli 1990). However, different studies showed that the influence of the uppermost k values of the boreholes as we used in this study are not of strong importance (Marmy et al. 2016, Pellet et al. 2017). Therefore, the used value is valid also for the other sites.*

*We used the upper most two thermistors at each borehole for our calculations.*

*References:*

*Vonder Mühll, D. and Haeberli, W., 1990. Thermal characteristics of the permafrost within an active rock glacier (Murtèl/Corvatsch, Grisons, Swiss Alps). Journal of Glaciology, 36(123): 151-158.*
*Marmy, A., Rajczak, J., Delaloye, R., Hilbich, C., Hoelzle, M., Kotlarski, S., Lambiel, C., Noetzli, J., Phillips, M., Salzmann, N., Staub, B. and Hauck, C., 2016. Semi-automated calibration method for modelling of mountain permafrost evolution in Switzerland. The Cryosphere, 10(6): 2693-2719.*
*Pellet, C. and Hauck, C., 2017. Monitoring soil moisture from middle to high elevation in Switzerland: Set-up and first results from the SOMOMOUNT network. Hydrological Earth System Science, 21: 3199-3220.*

p8 l24-26: will it make sense to add, for example in table 2, a column indicating a summary of missing data (% of NA) for all variables at the 3 sites?

*Percentage of remaining hourly data gaps are given for level 1 in chapter 3.2 Data Processing only for air temperature and snow height. The other data gaps can be investigated by the users from the provided PERMOS data.*

p9 l2-4: What does this mean? variable1@site1 is gapfilled using correlations like variable1@site1 ~ variable2@site1? "Energy balance variables" means also gapfilling wind speed with air temperature? can you clarify this? The results of this procedure are not included in table 2 right? Is there any reference to this in section 3.2?

*We only took data for the gap filling that have some physical relations to each other - these variables are shown in table 2. For example, we never used air temperature to gap-fill wind speed data. Wind speed was only gap-filled at Schilthorn (also mentioned in table 2) where a meteorological station of the IMIS network that includes wind speed measurements is situated very close on the top of the Schilthorn summit. These data were used to gap-fill some*

*of our values at the measurement station. At the other two stations, no meteorological stations are available nearby that could be used to gap-fill wind speed measurements. There is a reference to table 2 in the Data Processing 3.2 chapter.*

p9 l13: "Snow height data was" ... isn't "data" always plural? -> were corrected?

*Thanks for this comment, we had a mix of singular and plural in our paper. For data, actually, both singular or plural can be used (see e.g. definition by Oxford English dictionary). We homogenized all occurrences of the word data in our paper set it to plural!*

p9 l16: which "top of atmosphere radiation value" was used?

*With top of atmosphere radiation value, we mean 1366 W/m^2. We added this value in the sentence.*

p9 l21: "In addition, site specific processing steps were performed for the different variables." that are the ones presented @l22 and l28?

*Yes, these are the ones presented.*

p9 l23-25: snow height multiplication factors: all factors are <1. Does this mean that at Schiltorn, Stockhorn, and Murtel snow height is always lower than corresponding reference stations or vice versa? Can you comment on this? Wind erosion? Secondly, correction factors were used to gapfill missing snow data at the three sites or to systematically correct all snow data at the three sites? I imagine the first one. right? maybe you can specify it.

*Yes, in general this is right. However, in detail (daily measurements) it can change considerably as snow has of course a very high spatial and temporal variability but over longer periods the used calibration factors showed a very high and good correlation with the snow heights at the stations. We modified the sentences accordingly.*

p9 l28: "Level 2 data were ..." -> "Level 2 shortwave incoming radiation data were corrected ... with shortwave outgoing radiation" Am I right? Secondly, I imagine this procedure was adopted to correct QSin data during or right after snowfalls. Correct? How did you flag snowfall events and how did you define the time window when the correction is needed (how many hours after snowfall)? Did you find a nice way to flag the records when the QSin sensor is "obscured" by snow and thus the correction needs to be applied?. Lastly, I imagine alpha stands for albedo.

*Yes, your first comment here is absolutely right. Secondly, the pyranometers were neither heated nor ventilated; we tested if the station has snow cover according to snow height measurements at the station. Sometimes, snow covered the sensors for a couple of days. If the sensor is covered by snow, Qswin is smaller than Qswout. If this was the case, we corrected Qswin using alpha for fresh snow (alpha=0.87). Thirdly, yes alpha stands for albedo. We modified the sentences accordingly to better explain our procedure.*

p9 l29: Longwave incoming radiation correction: systematically? all data?

*Yes, all data were corrected.*

p10 l7 "available" .. what do you mean by that? originally available or after gapfilling? are there remaining gaps after the gapfilling procedure presented at p9? if yes it could be relevant to insert this information somewhere (see comment p8 l24-26)

*Finally, all the data we presented in our study are not completely gap filled even after the gap filling procedure (see comment on p8 l24-26 as you mention). After the gap filling procedure, the least gaps are present in air temperature and snow cover data. All other data may still have some longer gaps even after the gap filling procedure. Percentage of remaining hourly data gaps are given for level 1 in chapter 3.2 Data Processing only for air temperature and snow height. The other data gaps can be investigated by the users from the provided PERMOS data.*

p10 l22 and p12 l6-l14: how was the warming rate computed? I recommend the use of Mann–Kendall nonparametric trend test and Sen-slope estimator of trend.

*We recalculated the warming rates using the Mann-Kendall test (and then using Sen-slope values) (i) over the entire observation period of each site and (ii) over the same observation periods (2003-2018):*

*(i) : Entire observation (which differs from site to site) period warming rates*

- *Schilthorn: + 0.054 °C/year , p > 0.05*
- *Stockhorn: + 0.036 °C/year, p > 0.05*
- *Murtel:  +0.038 °C/year , p > 0.05*

*(ii) Warming rates over the same observation periods (2003 – 2018)*

- *Schilthorn: + 0.057 +C/year , p > 0.05*
- *Murtel:  +0.073 °C/year, p >0.05*

*We adapted the text in the manuscript accordingly.*

p10 l23: "for the periods in winter" what does this mean?

*'Periods in winter' means periods where/when snow cover is measured at our stations. A corresponding sentence was added to the text.*

p10 l25: "maximum snow height": is this the absolute maximum of the time series or the mean of yearly maximum values? I think that the latter is more informative.

*We mean here the absolute maximum ever measured during our measurement period. We find that both values are important. In the revised version we will also give the mean values over all measurement periods. The mean snow height of the yearly maximum values are:*

- *Stockhorn: 0.31 m*
- *Schilthorn: 0.88 m*
- *Murtel: 0.53 m*
- *We adapted the text in the manuscript accordingly.*

p10 l25: increasing and decreasing snow height trends are significant? see the previous comment regarding trend estimation

*Yes, we fully agree with you that snow height trends are very significant for the ground thermal regime. We have worked particularly hard to make the snow height information in our data as good as possible, also for further studies working with our data. We took your suggestion into account and estimated the trends using Mann-Kendall and Sen-slope. P-values are for all sites > 0.05, thus, not statistically significant:*

- *Stockhorn: +0.013 m/year, p > 0.05*
- *Schilthorn: - 0.05 m/year, p > 0.05*
- *Murtel: + 0.025 m/year, p >0.05*

We adapted the text in the manuscript accordingly.

p10 fig3: also the significance of these trends needs to be tested with Mann–Kendall nonparametric trend test and Sen-slope estimator of slopes of trend. Modify fig3 accordingly inserting p values and some measure of uncertainty around the trend values

We added p-values, calculated with Mann-Kendall test, to the plots.

p11 l7: albedo values: are mean albedo values computed using all days? (both snow and snow-free periods)? Are these differences caused by differences in snow cover duration or driven by albedo differences of the ground/rock surface spectral properties related to granulometry and/or lithology?

*Yes, the values are mean values for all days for snow covered and snow-free periods.*

p11 l9: "... the following values were measured ..." maybe " ... the following mean values were observed ... "

*We have corrected 'measured' to 'observed'.*

p11 l13-14: radiation components trends: see previous comments on significance.

*We also did Mann-Kendall test on these data and added the p-values to the plots.*

p11 l25 and l28 fig6 and fig7 legend: use the same notation used in fla1: Qs, Qr ...

*We homogenized the notations in the figure legends and the text. Thanks for pointing this out.*

p11 l26 "The share" maybe "the partitioning"?

*Yes, we corrected 'the share' to 'the partitioning'*

p11 l30: influence of snow cover on energy fluxes is more than "also" of importance. I think it is the main driver of the seasonal course you are describing.

*Yes, you are right. We changed the sentence to 'is of very high importance' and deleted 'also'*

p11 l31: "... snow is impacting the incoming shortwave radiation by its high albedo ... " snow albedo impacts net radiation rather than incoming shortwave radiation.

*We wanted to explain that in springtime, it is crucial whether we have a snow cover on the ground in comparison to having no snow cover. If we have no snow cover all the incoming energy of the incoming shortwave radiation is used directly to heat the ground surface whereas with a snow cover first the snow must be melted and takes away this shortwave energy.*
*Therefore, we reformulated the sentence in the following way:*
*"Particularly in spring, snow is consuming the entire energy of the incoming shortwave radiation to be melted and large differences are occurring in the energy transfer if the ground is snow covered or not."*

p11 l32 "available atmospheric energy" why not simply "available energy"?

*We corrected 'available atmospheric energy' to 'available energy'*

p12 l3: typo: (r−2)

*We corrected this typo.*

p12 l13: tpyo ... by (MeteoSwiss)

*We corrected this typo.*

p12 l16: "increase in cloudiness". If the significance of radiation trends is confirmed (see comment p11 l13-14), it is a very interesting point strongly related to the EDW discussion @p13 l3-l10. Do you have any other references or observations available? (MeteoSwiss data?). Can you exclude instrumental drift?

*This is a very good question. However, we cannot really answer it in detail as we do not have very good and long-term data at hand. However, some studies like Sanchez-Lorenzo and Wild (2012) report that during the three decades from 1981 to 2010 the estimated clear-sky surface solar radiation (SSR) trends reported in their study are in line with previous findings over Switzerland based on direct radiative flux measurements. They explain in their paper that the estimated all-sky SSR trends show a general agreement with cloud cover variability before the 1980s over Switzerland; at this point, a discrepancy in the sign of the trend is visible in the series. They also state that a dimming is clearly visible in all-sky SSR during the 1950s–1970s and afterwards a brightening is visible for 1980s–2000s period but this particularly also for lower altitude stations. Another study by Nyeki et al. 2019 show that the trends of meteorological parameters and surface downward shortwave radiation (DSR) and downward longwave radiation (DLR) were analysed at four stations with altitudes between 370 and 3580 m a.s.l. in Switzerland for the 1996–2015 period. An interesting point in this study is that the authors show for three of the four station an increase in DSR for three of the four stations (like the study of Sanchez-Lorenzo and Wild 2012) but at decrease one station. This station which showed a decrease was, however, the Jungfraujoch station high altitude measurement site at 3500 m a.s.l. This would actually fit with our observation, particularly for Schilthorn which is a mountain summit very close situated to Jungfraujoch. However, they also mention in their paper that these values are non-significant.*

*Related to the question about instrumental drift, we had a conversation with Laurent Vueilleumier from MeteoSwiss (specialist for radiation measurements) and he told us from his long measurement experience that if we used at our investigation sites always the same sensors at our investigation sites, in general the accuracy is much better in general than if we would have changed the sensors each year to adjust their calibration to the newest technology. Therefore, we are currently somehow sure that the measurements may have a certain drift but that this trend in general is probably smaller than the observed real trends.*

*References:*

*Sanchez-Lorenzo, A. and Wild, M., 2012. Decadal variations in estimated surface solar radiation over Switzerland since the late 19th century. Atmos. Chem. Phys., 12(18): 8635-8644.*
*Nyeki, S., Wacker, S., Aebi, C., Gröbner, J., Martucci, G. and Vuilleumier, L., 2019. Trends in surface radiation and cloud radiative effect at four Swiss sites for the 1996–2015 period. Atmos. Chem. Phys., 19(20): 13227-13241.*

p12 l18 "... the strong spatial differences between them are significant." This part of the sentence is unclear.

*We changed this sentence: ...the large topographical differences between them are considerable.*

p12 l24-26 and fig8 I recommend the use of Mann–Kendall nonparametric trend test and Sen-slope estimator of trend rather than linear regression.

*The Figures and manuscript text was changed accordingly (see answers above)*

p14 l11 "The data is available on different data platforms". Isn't this a repetition of l9?

*Yes, this is true. We removed this sentence.*

p12 l21: typo "... see (Scherler ..."

*Yes, we corrected this typo.*

p26 fig8, p27 fig9, p28 fig10 and p 29 fig11 captions: check typos sites-dates ()

*Yes, we corrected these typos.*

p32 table3: second-last row. typo "height"

*Yes, we corrected this typo.*